# The Synthesis and Performance of a Novel Lignin Modified Salt-Resistant Branched High-Performance Water Reducer

**DOI:** 10.3390/polym16020204

**Published:** 2024-01-10

**Authors:** Haipeng Xin, Donggang Guo

**Affiliations:** 1Institute of Environmental Science, Shanxi University, Taiyuan 030006, China; 2College of Environment and Resource, Shanxi University, Taiyuan 030006, China; gdghjkx@126.com; 3Shanxi Laboratory for Yellow River, Taiyuan 030006, China

**Keywords:** water reducer, salt resistance, dispersant

## Abstract

A lignin modified salt-resistant branched high-performance water reducer was prepared via free radical polymerization. The water-reducing agent was identified through its NMR spectrum, elemental analysis, Fourier transform infrared analysis, thermal gravimetric analysis, and scanning electron microscopy. The experiment conducted on cement paste demonstrates that the water-reducing efficiency can reach a maximum of 44%. Additionally, the significant spatial steric hindrance of the application enhances the dispersal capability of the water-reducing agent, resulting in effective water reduction and reduced viscosity. In addition, its compressive strength is the highest after 3-day curing and 3-, 7-, 28-day standard curing, and it has the best overall performance both in water and saline water prepared systems. The application in oil cement slurry shows that it exhibits a good dispersibility in fresh water, saline water, and substitute ocean water. In the Halfaya and Missan Oilfields of Iraq, BHPWR was used in a slurry with a density of 2.28 g/cm^3^ for casing the salt paste layer of five wells. The cementing results exceeded expectations with 100% qualified including over 85% excellent.

## 1. Introduction

The water reducer is a crucial admixture in the concrete industry, which has witnessed a shift from the conventional polyelectrolyte water reducer to the polycarboxylic acid water reducer [1,2,3,4]. The comb type polycarboxylic acid water reducer, characterized by its remarkable adjustability and significant potential for achieving a high performance [5,6,7], effectively addresses concerns related to dispersibility and stability, thereby yielding exceptional application outcomes [8,9,10].

However, it is impossible to directly use seawater, saline water, and alkali water to build coastal wave-resistant dikes and saltwater lake dams due to the decreased flow ability induced by salt resistance, which requires the long-distance transportation of fresh water, increasing costs and fuel consumption during transportation.

A salt-resistant water reducer could effectively increase the flow ability of concrete slurry containing salt which could help in the direct preparation of slurry with seawater, saline water, and alkali water.

This study is still in its early stages, and there is no relevant research yet. The available salt-resistant water reducer literature is mostly about the resistance of erosion of concrete by sulfate and chloride salts after being added [11,12,13,14,15,16,17,18], rather than to improve its fluidity during construction. 

Lignin and its derivates are the most abundant biomass resource due to lignin’s impressive characteristics like its high abundance, high molecular weight, oxidation resistance, bacteriostasis, and biodegradability, and they have been widely used for water treatment [19,20], as adhesive-reinforcing agents [21], water reducers [22,23], and even in hydrogen production [24].

Lignosulfonate is the predominant biomass water reducer in use currently. Although its shortcomings, such as a low water-reducing efficiency and high level of air entrainment and retardation, constrain its broader application, lignin and lignosulfonate still carry the advantages of being environmentally friendly and cost-effective. 

In this work, a salt-resistant branched high-performance water reducer (BHPWR) was prepared by introducing branching structures starting from the multi-hydroxyl lignin, The main chain of BHPWR contained carboxyl and sulfonic acid groups as well as long hydrophilic side chains. By utilizing the strong steric hindrance effect of lignin and polyethylene glycol monocetyl ether in the side chains, the volume dispersion and electrostatic repulsion led to strong dispersing effect which could effectively improve concrete fluidity under saline conditions. The BHPWR was experimented with by chemical analyses such as ^1^H-NMR, Fourier transform infrared, elemental analysis, thermogravimetric analysis, and scanning electron microscope, as well as performance analysis in concrete and cement slurry, which indicated that BHPWER performs good water-reducing rate in concrete prepared with saline water and good dispersion in cement slurry prepared by saline water. And the BHPWR-3 was used to cement a 244.5 mm salt paste layer casing for five wells in the Halfaya and Missan Oilfields in Iraq.

## 2. Materials and Methods

### 2.1. Materials

Acrylic acid (AA) was obtained from Sinopharm Chemical Reagent Co., Ltd., Shanghai, China. AMPS was obtained from Nanjing All-Plus Chemical Co., Ltd., Nanjing, China. APEG-2000 was obtained from Haian Petrochemical Plant Co., Ltd., Haian, China. Lignin was obtained from Shandong Gaotang Polymeric Lignin Co., Ltd., Liaocheng, China. Ammonium persulfate (APS) was obtained from Shandong Zhengxing New Material Co., Ltd., Zibo, China. Tetramethylethylenediamine (TMEDA) was obtained from Shandong Zhenkun New Material Co., Ltd., Liaocheng, China. Ethanol was obtained from Xi’an Dongsen Chemical Material Co., Ltd., Xi’an, China.

P·O 52.5 Cement was obtained from Jidong Cement Co., Ltd., Tangshan, China. Quartz sand ranging from 20 to 120 mesh was obtained from Hebei Huakai Mining Co., Ltd., Xingtai, China. Silica fume with a SiO_2_ content exceeding 94% and a specific surface area of 20 m^2^/g was obtained from Gansu Sanyuan Silicon Material Co., Ltd., Lanzhou, China. Class I fly ash with a density of 2200 kg/m^3^ was obtained from Shandong Shunke Building Materials Technology Co., Ltd., Yantai, China. Glass beads with a density of 412 kg/m^3^ and a particle size of approximately 0.1 mm were obtained from Xingtai Guangqing Refractory Material Distribution Co., Ltd., Xingtai, China. Steel fiber with a diameter of 0.22 mm and length ranging from 13 to 15 mm was obtained from Changzhou Tianyi Engineering Fiber Co., Ltd., Changzhou, China. Defoaming agent was obtained from Jiangsu Liqi Environmental Technology Co., Ltd., Yancheng, China. NaCl was obtained from Tianjin Changlu Haijing Group Co., Ltd., Tianjin, China.

All reagents were of industrial grade and were used without further purification. The water used was distilled three times.

### 2.2. Characterization

The ^1^H-NMR experiment was performed on a Bruker AVANCE400 NMR spectrometer, Billerica, MA, USA. D_2_O was used for field-frequency lock, and the observed ^1^H chemical shifts were reported in parts per million (ppm) relative to an internal standard (TMS, 0 ppm). 

The Fourier transform infrared (FTIR) experiment was performed on Bruker Tensor 27 spectrometer with BHPWR in KBr pellets at room temperature. The spectra were recorded in the range of 4000–400 cm^−1^.

Elemental analysis was performed on an Elementar Vario E1 III analyzer, Rhine Main, Germany.

Thermogravimetric analysis (TGA) of the dried sample was performed using a Mettler Toledo TGA/DSC 1, Zurich, Switzerland, in the temperature range 30–700 °C with a heating rate of 10 °C/min under a nitrogen atmosphere.

The scanning electron microscope (SEM) measurement was obtained using an JEOL JSM-7600 Thermal Field Emission Scanning Electron Microscope, Akishima, Japan, and the surface of the polymer was treated with gold spraying as previously.

The water reduction rate was tested according to GB 8076-2008 [25]. The concrete mix ratio was: m (Jidong Cement): m (sand): m (stone) = 360: 810: 990.

Concrete viscosity was tested according to T_500_ in JGJ/T 283-2012 [26].

The mechanical properties of concrete are tested in accordance with the GB/T 50081-2019 standard [27] through the process of standard steam curing. The specimens, after being exposed to an experimental setting for 24 h at a temperature of (20 ± 5) °C and a relative humidity above 50%, are transferred to a curing box. They are then gradually heated, not surpassing a rate of 15 °C/h, until reaching a temperature of (90 ± 1) °C. The specimens are maintained at a constant temperature for 48 h before being cooled down, again not exceeding a rate of 15 °C/h, until reaching (20 ± 5) °C.

The process of curing at the typical temperature of a room is conducted in accordance with the guidelines provided by GB/T 50081-2019 [27], which outlines the standard procedures for evaluating the mechanical properties of concrete. Samples are placed in a typical curing chamber at a temperature of (20 ± 2) °C and a relative humidity exceeding 95%.

A ZNN-D6 rotational viscometer was used to conduct the rheological test at 85 °C and atmospheric pressure, following the guidelines of SY/T 5504.3-2018 [28] “Evaluation Method for Oil Well Cement Additives Part 3 Friction Reducers”.

## 3. Synthesis and Characterization of BHPWR

### 3.1. The Synthesis of BHPWR

One linear water reducer (LWR) and six BHPWR samples were synthesized via free radical polymerization as shown in Figure 1. The compounds 2-acrylamide-2-methylpropanesulfonic acid (AMPS), acrylic acid (AA), allyl polyethylene glycol with molecular weight of 2000 g/mol (APEG-2000), lignin, and water were added in one portion and stirred to dissolve at 70 °C forming a solution with lignin and 125 g monomers and 375 g water, then the initiators APS (0.5 g, 0.0022 mol) and TMEDA (0.26 g, 0.0022 mol) were added, and the solution was kept at 70 °C for 3 h. The resulting pale yellow liquid was filtered while still hot, then precipitated by ethanol, and washed by ethanol 6 times and a mixture of water/ethanol (3/7, *v*/*v*) 3 times to remove unreacted monomers. The products were dried in a vacuum at 70 °C for 48 h. 

The composition of samples was determined by the elemental analysis. The weight percentages of carbon, hydrogen, nitrogen, and sulfur from elemental analyses of the samples could be obtained. However, we can only give the actual mole ratio of AMPS in polymer via the weight percentages of sulfur as listed in Table 1. The results indicate that the reactivity ratio of AMPS is less than that of AA and APEG-2000 in the copolymerization.

### 3.2. The Characterization of BHPWR

The observed ^1^H chemical shifts of BHPWR-5 were reported as shown in Figure 2, (a) –CH_3_ of AMPS and lignin appeared at 1.2–1.3 ppm, (b) –CH_2_ of AA, APEG-2000, and AMPS in the polymer backbone and lignin appeared at 1.3–1.6 ppm, (c) -CH of AA, APEG-2000, and AMPS in the polymer backbone, and Ar-CH_3_ lignin appeared at 1.9–2.3 ppm, (d) –CH_2_ of APEG-2000 on the side chain, on the AMPS nearing -SO_3_H, and –O–CH_2_ and –OH on lignin appeared at 3.0–4.1 ppm, (e) the double bond (–CH_2_=CH–) of unreacted monomers and –CH_2_=CH– of the benzene ring on lignin appeared at 5.1–6.7 ppm.

The IR spectrum of BHPWR-5 is presented in Figure 3 with the main absorption bands of BHPWR-6 assigned as follows: (a) OH stretching (H_2_O, APEG-2000) appeared at 3500 cm^−1^, (b) CH_2_ and CH_3_ stretching (AMPS, APEG-2000, lignin) appeared at 2880 cm^−1^, (c) C=O symmetric stretching (acrylic acid) appeared at 1720 cm^−1^, (d) aromatic ring skeleton stretching (lignin) appeared at 1653 cm^−1^ and 1467 cm^−1^, (e) COO-symmetric stretching (acrylic acid) appeared at 1345 cm^−1^, (f) C–O–C stretching (APEG-2000, lignin) appeared at 1114 cm^−1^, and (g) S–O bending (sulfonic acid group of AMPS) appeared at 626 cm^−1^.

TGA and DTG curves of BHPWR-5 are shown in Figure 4. The thermogram of BHPWR-5 showed three steps during its mass loss. The first stage of 16.0% weight loss was related to the thermal decomposition of the polyethylene glycol monocetyl ether at 234–382 °C. The second stage of 36.1% weight loss was attributed to the thermal decomposition of the ether bond formed by the –OH of lignin and –CH=CH_2_ of monomers at 380–430 °C. The third step of 29.8% weight loss may be attributed to polymer backbone degradation at 430–470 °C. The results indicate that BHPWR exhibits a good temperature resistance. 

An electron microscopy test was conducted to further study the stereostructure of the water reducer in solution, using scanning electron microscopy (SEM). BHPWR-5 did not exhibit a single regular shape due to the free rotation of its main chain and side chains. Despite the adsorption of the primary chain onto the cement particle surface in the cement slurry, the secondary chain exhibits erratic movement as well. Figure 5 shows the aggregate state of the water reducer at the micrometer scale, which is a dendritic shape. The primary chain of the polymer attaches to the cement particles in the cement slurry through carboxyl and sulfonic acid groups, while the polymer’s secondary chains are long side chains that can extend without restriction when the primary chain is attached to the cement slurry. The response of water and the mechanical radius increases due to the significant molecular weight of the side chains, resulting in a substantial spatial steric hindrance. Due to this large spatial steric hindrance effect, the side chains have a strong three-dimensional dispersing ability on cement particles, which leads to a good dispersion of cement slurry and further increases its rheology.

## 4. Concrete and Cement Slurry Performance 

### 4.1. Concrete Performance 

Table 2 and Table 3 display the performance of concrete made with freshwater and saltwater, correspondingly. The water-reducing rate of BHPWR-3 could reach up to 45% in fresh water and 44% in saline water, and its large spatial steric hindrance improves the dispersibility of the water reducer, achieving a high water reduction and flow rate which help to reach a 500 mm slump quickly. 

In addition, its compressive strength is the highest after 3-day curing and 3-, 7-, and 28-day standard curing, and it has the best overall performance both in fresh water and saline water prepared systems.

The scheme model of water reducers compressed by salt is given in Figure 6. Chains of LWR were compressed and the gyration radius of copolymer coils decreased sharply from the stretched conformation in aqueous solution when salt ions were added. The small gyration radius of copolymer coils could not disperse the cement particles. Thus, LWR could not reduce water in saline water with a high salt content. Chains of BHPWR-0, which has a long side chain but no lignin, were also compressed and the gyration radius of copolymer coils decreased sharply, less sharply than LWR, from the stretched conformation in aqueous solution when salt ions were added. The gyration radius of copolymer coils, which is bigger than LWR due to the three-dimensional dispersion effects of side chains, could partly disperse the cement particle. Thus, BHPWR-0 could reduce some water in saline water with a high salt content. When it comes to water reducers with APEG-2400 and lignin (BHPWR-1, BHPWR-2, BHPWR-3, BHPWR-4, BHPWR-5), the chains were also compressed and the gyration radius of copolymer coils decreased. However, the gyration radius of copolymer coils with side chains and a lignin structure could still maintain a larger size due to the three-dimensional dispersion effects of the side chains and impregnable lignin structure. Thus, they could effectively reduce water in saline water with a high salt content.

### 4.2. Cement Slurry Performance 

BHPWR could also be used as oil cement dispersant. BHPWR-3 was added to the slurry prepared with fresh water, saline water with 36% NaCl by weight of water, and substitute ocean water which is prepared according to Designation D1141-98 (D1141-98). The dispersant underwent evaluation in accordance with SY/T 5504.3-2008 and the important indicators are the flow behavior index *n* and consistency index *K* calculated from *θ*_300_ and *θ*_300_, the reading of viscometer at the shearing rate of 100 r/min and 300 r/min. A good dispersant should result in a bigger *n* and lower *k*.
(1)$$n=2.096\times lg\frac{\theta_{300}}{\theta_{100}}$$
(2)$$K=\frac{0.511\times \theta_{300}}{511^{n}}$$

The rheology of slurries prepared with fresh water, saline water, and substitute ocean water is listed in Table 4. With the addition of 1.5% by weight of cement (BWOC), *n* = 0.78–0.93 and *K* = 0.04–0.16 Pa.s^n^ at 52 °C and atmospheric pressure, while *n* = 0.73–0.92 and *K* = 0.16–0.19 Pa.s^n^ at 85 °C and atmospheric pressure, which indicates that BHPWR-3 exhibits a good dispersibility in fresh water, saline water, and substitute ocean water.

## 5. The Application of BHPWR

The cement slurries used in Iraq’s Halfaya and Missan Oilfields were prepared by saline water with a low fluidity, significant thixotropy, and high initial consistency due to its high solid and salt content. Furthermore, the elevated temperature during the summer in Iraq speeds up the process of cement hydration, ultimately causing a decrease in the rheology of the cement slurry. Due to the persistent high ground temperature, which can reach up to 60 °C during the summer in Iraq, the cement slurry experienced significant hydration while being mixed, leading to a decrease in its fluidity as shown in Table 5 which shows *θ*_300_ unreadable at 60 °C, strong thixotropy, and a high initial consistency leading to difficulty in pumping and unstable density which seriously impacts on on-site construction safety and cementing quality. The Sector Bond Tool (SBT) results showed that the salt paste layer casing in Halfaya and Missan Oilfields was only about 60% qualified including 20–30% excellent.

BHPWR could enhance the flow characteristics of slurry, making it easier to mix and place. Rheology is a metric that gauges the ability to resist fluid motion and the decrease in such resistance when subjected to pressure. Upon substitution of the dispersant in Table 6 with BHPWR-3, the slurry displayed favorable rheological characteristics, with a flow behavior index of *n* = 0.74 and a consistency factor of *K* = 1.08 Pa.s^n^ at 60 °C from which we can learn that BHPWR-3 represents good dispersant performance.

To cement the 244.5 mm salt paste layer casing for five wells in the Halfaya and Missan Oilfield in Iraq, the BHPWR-3 was utilized with high-density (2.28 g/cm^3^) saline cement slurry under the ambient temperature of 55–58 °C. One of the wells, namely FQCS-44H, serves as a prime example, involving the sealing of the Lower Fars formation containing high-pressure saltwater zones via casing measuring 244.5 mm in diameter placed in 311.1 mm open holes, with the shoe being at 2957.7 m as opposed to the previous 339.7 mm casing at 2081.5 m. However, it is challenging to conduct cementing under these circumstances given that a higher mud weight (2.23 g/cm^3^) requires a denser slurry, potentially narrowing the cementing window. Despite these complications, the SBT evaluation indicates that the cementing job was 100% qualified including over 85% excellent.

## 6. Conclusions

BHPWR prepared via free radical polymerization with a good temperature resistance, which decomposes from 234 °C, demonstrates an excellent performance as a water reducer, with reduction rates of up to 45% in fresh water and 44% in saline water. After 3-day curing and standard curing for 3, 7, and 28 days, it shows the highest compressive strength and overall best performance in both water and saline water systems. Furthermore, it has been found to exhibit a good dispersibility in fresh water, saline water, and substitute ocean water, making it a suitable dispersant for oil cement slurries. Rheological testing of slurries prepared with BHPWR-3 showed *n* = 0.78–0.93 and *K* = 0.04–0.16 Pa.s^n^ at 52 °C and atmospheric pressure, and *n* = 0.73–0.92 and *K* = 0.16–0.19 Pa.s^n^ at 85 °C and atmospheric pressure. This indicates that BHPWR-3 is effective in fresh water, saline water, and substitute ocean water. In the Halfaya and Missan Oilfields of Iraq, BHPWR was used in slurry with a density of 2.28 g/cm^3^ for casing the 244.5 mm salt paste layer of five wells. The cementing results exceeded expectations with 100% qualified including over 85% excellent.

## Figures and Tables

**Figure 1 polymers-16-00204-f001:**
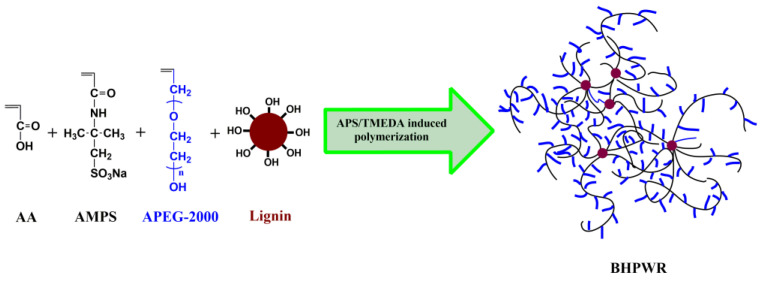
Synthesis route of BHPWR.

**Figure 2 polymers-16-00204-f002:**
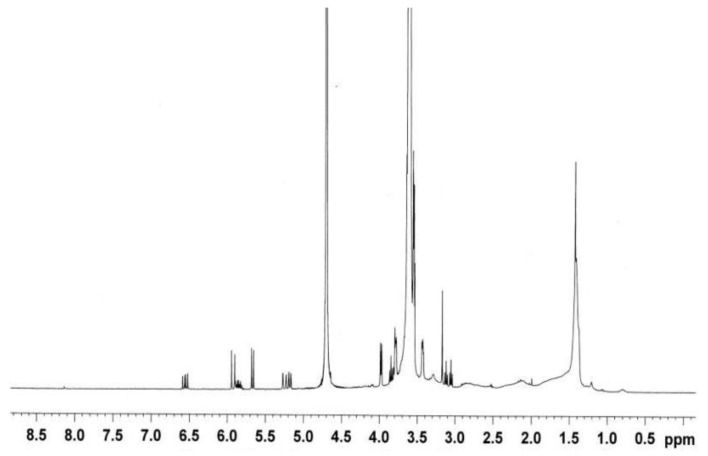
NMR spectrum of BHPWR-5.

**Figure 3 polymers-16-00204-f003:**
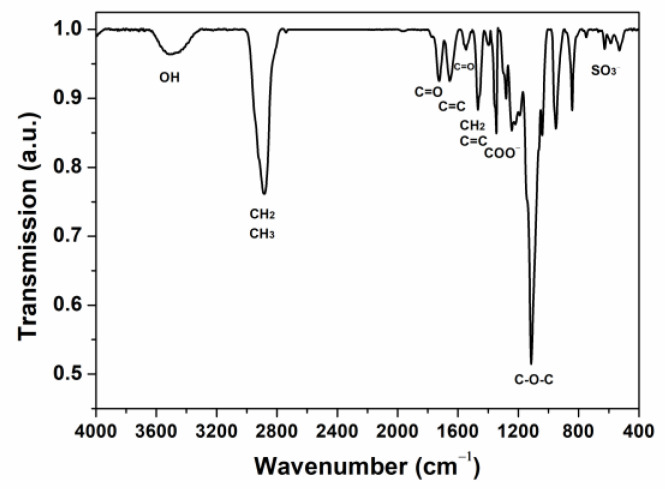
Infrared spectra of BHPWR-5.

**Figure 4 polymers-16-00204-f004:**
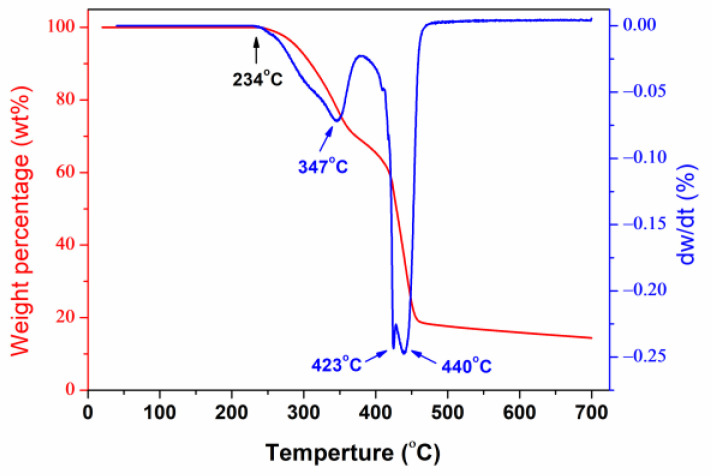
TGA and DTG curves of the BHPWR-5.

**Figure 5 polymers-16-00204-f005:**
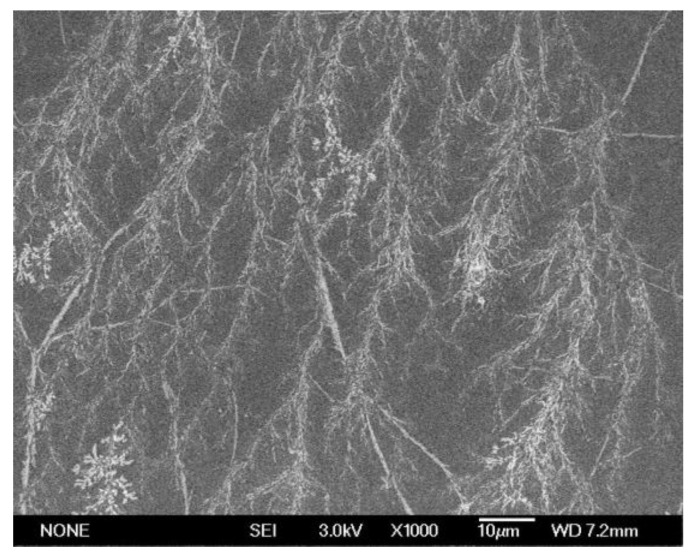
SEM of BHPWR-5.

**Figure 6 polymers-16-00204-f006:**
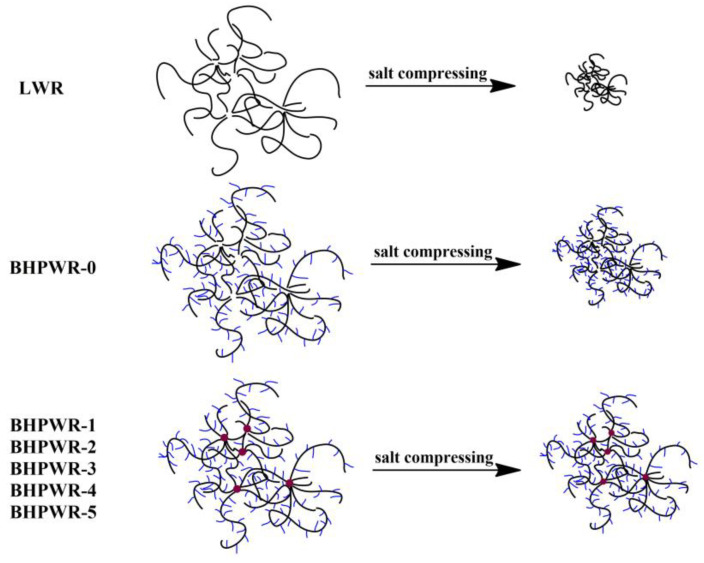
The scheme model of the water reducers compressed by salt.

**Table 1 polymers-16-00204-t001:** Characteristics of 6 samples.

Sample	AA ^1^ (mol%)	AMPS ^1^ (mol%)	APEG-2000 ^1^ (mol%)	Lignin g	AMPS ^2^ (mol%)	Yield %
LWR-0	30.4	69.6	0.7	0	68.4	93.4
BHPWR-0	30.2	69.1	0.7	0	67.2	93.1
BHPWR-1	30.2	69.1	0.7	0.5	67.1	94.0
BHPWR-2	30.2	69.1	0.7	0.7	66.4	93.5
BHPWR-3	30.2	69.1	0.7	0.9	65.3	92.6
BHPWR-4	30.2	69.1	0.7	1.1	65.0	90.8
BHPWR-5	30.2	69.1	0.7	1.3	65.1	91.6

^1^ mol% monomer present in the feed ratio. ^2^ Determined via elementary analysis.

**Table 2 polymers-16-00204-t002:** Performance of concrete prepared by fresh water ^1^.

Sample Added	Water Reduction Rate %	Concrete Slump mm	T_500_ s	3-Day Compressive Strength Under Steam Curing MPa	Compressive Strength MPa		
					3 d	7 d	28 d
LWR	15	572	8.46	65.7	45.7	61.1	66.3
BHPWR-0	28	684	7.39	97.6	64.9	90.6	93.5
BHPWR-1	36	766	6.62	120.1	83.1	115.4	118.6
BHPWR-2	41	799	6.13	135.2	85.7	120.3	135.7
BHPWR-3	45	832	4.85	147.6	94.3	129.9	144.6
BHPWR-4	32	763	5.27	127.1	79.8	110.2	129.3
BHPWR-5	35	771	5.96	119.3	81.2	112.7	130.4

^1^ Concrete batching: Cement 700 g + fly ash 105 g + silica fume 150 g + foaming agent 50 g + quartz sand 1200 g + water 170 g + steel fiber 150 g + water reducer 21.4 g.

**Table 3 polymers-16-00204-t003:** Performance of concrete prepared by saline water ^1^.

Sample Added	Water Reduction Rate %	Concrete Slump mm	T_500_ s	3-Day Compressive Strength Under Steam Curing MPa	Compressive Strength MPa		
					3 d	7 d	28 d
LWR ^2^	0	-	-	-	-	-	-
BHPWR-0 ^3^	12	487	-	-	-	-	-
BHPWR-1	30	650	6.97	81.3	44.6	62.1	84.2
BHPWR-2	39	710	6.44	97.5	48.9	69.7	93.1
BHPWR-3	44	812	4.91	123.6	70.2	100.2	125.7
BHPWR-4	29	702	5.83	100.4	50.3	71.3	99.8
BHPWR-5	28	680	6.22	90.2	52.7	73.2	98.9

^1^ Concrete batching: Cement 700 g + fly ash 105 g + silica fume 150 g + foaming agent 50 g + quartz sand 1200 g + water 170 g + NaCl 61.2 g + steel fiber 150 g + water reducer 21.4 g. ^2^ The concrete could not flow after prepared. ^3^ The concrete could flow but the concrete slump was less than 500 mm.

**Table 4 polymers-16-00204-t004:** Rheology of slurries prepared with different kinds of water ^1^.

Water Type	Addition of BHPWR-3 %BWOC	Temperature °C	θ_3_	θ_6_	θ_100_	θ_200_	θ_300_	*n*	*K* Pa.s^n^
Fresh water	0	52	14	20	68	81	92	0.28	8.45
Fresh water	1.5	52	--	--	6	10	15	0.83	0.04
Saline water	0	52	14	19	36	47	60	0.46	1.68
Saline water	1.5	52	3	4	9	16	25	0.93	0.04
Substitute ocean water	0	52	20	29	90	95	105	0.14	22.4
Substitute ocean water	1.5	52	6	9	17	26	40	0.78	0.16
Fresh water	0	85	12	17	111	133	143	0.23	17.3
Fresh water	1.5	85	--	--	4	7	11	0.92	0.18
Saline water	0	85	16	21	49	59	65	0.26	6.67
Saline water	1.5	85	5	6	13	22	29	0.73	0.16
Substitute ocean water	0	85	20	26	145	153	162	0.10	44.2
Substitute ocean water	1.5	85	8	13	16	25	36	0.74	0.19

^1^ Slurry formulation: G class cement + water + BHPWER with a water–solid ratio of 0.44.

**Table 5 polymers-16-00204-t005:** Rheology of 2.28 g/cm^3^ slurry ^1^at different ambient temperatures.

Ambient Temperature °C	θ_3_	θ_6_	θ_100_	θ_200_	θ_300_	*n*	*K* Pa.s^n^
27	3	10	79	138	195	0.82	0.59
40	16	20	88	158	212	0.8	0.74
50	17	28	108	162	235	0.71	1.45
60	19	25	138	200	--	--	--

^1^ Slurry formulation: Oman G class cement 450 g + hematite 300 g + manganese ore fines BH-WS 22.5 g + fluid loss additive BH-F101L 18 g + dispersant BH-D301L 15.7 g + retarder BZR101 0.1 g + defoamer XP-1 0.9 g + salt NaCl 40.5 g + water 214.7 g.

**Table 6 polymers-16-00204-t006:** Rheology of 2.28 g/cm^3^ slurry using BHPWR-3 ^1^ at different ambient temperatures.

Ambient Temperature °C	θ_3_	θ_6_	θ_100_	θ_200_	θ_300_	*n*	*K* Pa.s^n^
27	2	8	59	102	150	0.85	0.38
40	6	14	73	125	173	0.79	0.66
50	8	19	86	146	195	0.75	0.95
60	9	23	96	158	217	0.74	1.08

^1^ Slurry formulation: Oman G class cement 450 g + hematite 300 g + manganese ore fines BH-WS 22.5 g + fluid loss additive BH-F101L 18 g + dispersant BHPWR-3 15.7 g + retarder BZR101 0.1 g + defoamer XP-1 0.9 g + salt NaCl 40.5 g+ water 214.7 g.

## Data Availability

Data are contained within the article.

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
