# Peer review of "The Synthesis and Performance of a Novel Lignin Modified Salt-Resistant Branched High-Performance Water Reducer"

_polymers, 2024, doi:10.3390/polym16020204_

Round 1

Reviewer 1 Report

Comments and Suggestions for Authors

polymers-2802806 – The Synthesis and Performance of a Novel Lignin Modified Salt Resistance Branched High-performance Water Reducer.

1-    The introduction could be improved by referring some new refs such as https://doi.org/10.1016/j.indcrop.2023.116720

2-    The authors must apply additional analyses to the BHPWR sample, such as TGA or XRD.

3-    Analyses of data/results should be further elevated for Table 2.

4-    A comparison of the study with existing similar published results should be mentioned at appropriate place.

I recommend this work for publication after major revision.

Author Response

I am very much thankful to the reviewer for the deep and thorough review. I have revised my present research paper in the light of your useful suggestions and comments. I hope my revision has improved the paper to a level of your satisfaction.

 Comments 1. The introduction could be improved by referring some new refs such as https://doi.org/10.1016/j.indcrop.2023.116720

Response: New refs such as https://doi.org/10.1016/j.indcrop.2023.116720 are referred. Finally, 24 refs are referred,including 3 refs published in 2024, 8 refs published in 2023, and 3 refs published in 2022.

 Comments 2. The authors must apply additional analyses to the BHPWR sample, such as TGA or XRD.

Response: 1H-NMR, elemental analysis, and thermo-gravimetric analysis scanning are added.

Comments 3. Analyses of data/results should be further elevated for Table 2.

Response: After adding a new Table 1, “Table 2” becomes Table 3 now. And analyses of data/results is now further elevated as requirement. To better illustrate the principle, we also add a figure presenting a scheme model of the water reducers compressed by salt.

Comments 4. A comparison of the study with existing similar published results should be mentioned at appropriate place.

Response: Sorry that we can’t find existing similar published paper in this area. This study is still in its early stages, and there is no relevant research yet. The available salt-resistant water reducer literatures are mostly about resistance of erosion of concrete by sulfate and chloride salts after being added, rather than to improve its fluidity during construction.

Reviewer 2 Report

Comments and Suggestions for Authors

We will consider publishing your review entitled " The Synthesis and Performance of a Novel Lignin Modified  Salt Resistance Branched High-performance Water Reducer ". This paper is suitable for the Journal of Polymers after a revision. This manuscript has the following issues.

1-     Introduction should be modified in terms of application of nano materials The references should be updated.

2-     Abstract section should be fully supported by the summarized results.

3-      Prepare the comparing table to compare different materials in terms of physicochemical properties.

4-     Please illustrate the merits of method that apply in this work besides simple and inexpensive, contrasting to other methods.

5-     Synthesis section should be modified with more details.

6-     How about the yield of the synthesized samples?

7-     Porosity of synthesized polymer should be examined.

8-     It is suggested to clarify the assembling the integrated device with more details. 

Author Response

I am very much thankful to the reviewer for the deep and thorough review. I have revised my present research paper in the light of your useful suggestions and comments. I hope my revision has improved the paper to a level of your satisfaction.

Sorry that i am unable to attach figures here. Please read file attched to find figure 1 and figure 2, Thank you.

 Comments 1. Introduction should be modified in terms of application of nano materials The references should be updated.

Response: Introduction was rewritten and new references are updated. Finally, 24 references are referred, including 3 references published in 2024, 8 references published in 2023, and 3 references published in 2022.

 Comments 2. The authors must apply additional analyses to the BHPWR sample, such as TGA or XRD. Abstract section should be fully supported by the summarized results.

Response: Abstract section is fully supported by the summarized results now.

Comments 3. Prepare the comparing table to compare different materials in terms of physicochemical properties.

Response: This study is still in its early stages, and there is no relevant research yet. The available salt-resistant water reducer literatures are mostly about resistance of erosion of concrete by sulfate and chloride salts after being added, rather than to im-prove its fluidity during construction.

To compare different materials, we introduced a linear water reducer LWR as traditional water reducer and comb type polycarboxylic acid water reducer BHPWR-0. The Performance of concrete with different kind of water reducers prepared by fresh water and saline water were compared. We also add a figure presenting a scheme model of the water reducers compressed by salt for further elevated.

Comments 4. Please illustrate the merits of method that apply in this work besides simple and inexpensive, contrasting to other methods.

Response: We synthesized a novel polymer which could improve the concrete and cement slurry performance while prepared by saline water. We thought it would be innovative since that the available salt-resistant water reducer literatures are mostly about resistance of erosion of concrete by sulfate and chloride salts after being added, rather than to improve its fluidity during construction. It may provide a wat to directly use seawater, saline water, and alkali water to build coastal wave-resistant dikes and saltwater lake dams. Thus, we thought that the merit of this work may be the application.

Comments 5. Synthesis section should be modified with more details.

Response: This section was rewritten with more details. We also add a figure presenting the synthesis route of the water reducer.

Comments 6. How about the yield of the synthesized samples?

Response: The yield of the synthesized samples was added in a new Table 1.

Comments 7. Porosity of synthesized polymer should be examined.

Response: We didn’t exam the porosity of synthesized polymers due to that they are not porous hydrogels but branched polymers as figure 1 and figure 2. There are neither pore in the polymers themselves nor inducing pores in concrete or cement slurry.

As the synthesis route we give in figure 1, APEG-2400 just provide a secondary branching structure while lignin proved the primary branching structure.

When we take APEG-2000 away from figure 1, we could also understand the structure of BHPWR as figure 2.

(Please read file attched to find figure 1 and figure 2)

Comments 8. It is suggested to clarify the assembling the integrated device with more details.

Response: Sorry that I can’t follow the meaning of assembling the integrated device.

We present a scheme model of the water reducers compressed by salt to make sure that the discuss of samples performance in saline was got further elevated.

Round 2

Reviewer 1 Report

Comments and Suggestions for Authors

After this revision, it in the manuscript ID "polymers-2802806" this paper is accept in present form and can be a very good contribution to polymers.